# Nerve Growth Factor Induces Proliferation and Aggressiveness in Prostate Cancer Cells

**DOI:** 10.3390/cancers11060784

**Published:** 2019-06-06

**Authors:** Marzia Di Donato, Gustavo Cernera, Antimo Migliaccio, Gabriella Castoria

**Affiliations:** Department of Precision Medicine-University of Campania ‘L. Vanvitelli’-via L. De Crecchio, 7-80138 Naples, Italy; marzia.didonato@unicampania.it (M.D.D.); gucern@yahoo.it (G.C.)

**Keywords:** NGF/TrkA signaling, mitogenesis, invasiveness, EMT, 3D models, castrate-resistant prostate cancers

## Abstract

Resistance to hormone therapy and disease progression is the major challenge in clinical management of prostate cancer (PC). Drugs currently used in PC therapy initially show a potent antitumor effects, but PC gradually develops resistance, relapses and spreads. Most patients who fail primary therapy and have recurrences eventually develop castration-resistant prostate cancer (CRPC), which is almost incurable. The nerve growth factor (NGF) acts on a variety of non-neuronal cells by activating the NGF tyrosine-kinase receptor, tropomyosin receptor kinase A (TrkA). NGF signaling is deregulated in PC. In androgen-dependent PC cells, TrkA mediates the proliferative action of NGF through its crosstalk with the androgen receptor (AR). Epithelial PC cells, however, acquire the ability to express NGF and TrkA, as the disease progresses, indicating a role for NGF/TrkA axis in PC progression and androgen-resistance. We here report that once activated by NGF, TrkA mediates proliferation, invasiveness and epithelial-mesenchymal transition (EMT) in various CRPC cells. NGF promotes organoid growth in 3D models of CRPC cells, and specific inhibition of TrkA impairs all these responses. Thus TrkA represents a new biomarker to target in CRPC.

## 1. Introduction

Prostate cancer (PC) is the most commonly diagnosed non-cutaneous tumor in men and the second leading cause of male cancer-related deaths in Western society [1]. This cancer is often characterized by a slow and symptom-free growth, and early-stage treatments include radical prostatectomy, novel androgen receptor (AR) pathway inhibitors, such as abiraterone and enzalutamide, external beam radiotherapy (RT), brachytherapy and cryotherapy [2,3]. Further, use of new tracers for Positron Emission Tomography/Computed Tomography (PET/TC) [4,5,6,7] and novel focal therapies [8,9] have provided important advances in diagnosis, follow-up and treatment of PC patients in the last decade. 

The role of steroid hormones, mainly the androgens, in prostate transformation and PC progression is well established and molecular studies have extensively analyzed the mechanism of AR or estrogen receptors (ERs) action in PC. These studies have made possible the synthesis of new drugs that modulate or inhibit the biological events induced by sex steroid receptors (SRs) in PC cells [10,11,12,13,14,15]. The long-term efficacy of these drugs, however, still remains unsatisfactory and novel therapeutic approaches are needed to limit PC progression and inhibit metastasis occurrence. 

Receptor tyrosine kinases (RTKs) drive prostate transformation and PC progression [16]. Among them, the tropomyosin receptor kinase A (TrkA) binds nerve growth factor (NGF) thereby activating Ras/mitogen-activated protein kinase (MAPK), phosphoinositide 3-kinases (PI3-K) and phospholipase C gamma (PLCγ) signaling pathways to promote survival, proliferation and invasiveness of cells [17,18]. The human prostate releases abundant levels of NGF [19], which, in turn, controls the normal development of prostate tissue [20]. Specifically, stromal cells secrete NGF, which binds to TrkA and p75NTR expressed in the epithelial counterpart, stimulating its growth [21,22,23]. Moreover, preclinical studies have highlighted the role of NGF/TrkA signaling in PC proliferation and metastasis [24,25,26,27,28,29,30]. Molecular changes in epithelial or stromal cells lead to the paracrine and/or autocrine production of NGF, thus promoting prostate carcinogenesis. Simultaneously, persistent expression of TrkA, together with the loss of expression of p75NTR receptors, is often detected in PC patients [26]. As such, PC cells might exclusively utilize NGF signaling to survive. Noticeably, we have recently shown that crosstalk between TrkA and AR modulates the NGF action in quite divergent cell types. This interaction controls the androgen-elicited differentiation in neuronal-derived cells [31], while it controls the NGF-induced proliferation and migration in androgen-sensitive LNCaP cells [32]. Therefore, TrkA represents a promising new ‘druggable’ biomarker in prostate proliferative diseases [33].

Despite the accumulating evidence, the mechanism(s) leading to derangement of TrkA signalling in castration-resistant prostate cancer (CRPC) remains, however, poorly understood and genetic screening did not reveal TrkA mutations or Trk-fusion onco-proteins in PC patients [34,35]. These findings further reinforce the concept that derangement of an intact NGF/TrkA signaling may be involved in PC progression.

In this study, we have used C4-2B [36,37], PC3 [38] and DU145 [39] cell lines, which derive from CRPC and express TrkA at varying levels [40]. C4-2B cells are also positive for AR [41], while PC3 and DU145 cells are both negative for AR [42,43]. Additionally, all the cell lines express ERβ, while they do not express ERα. In these cell lines, we have analyzed the effects of NGF, alone or in combination with its specific inhibitor GW441756 [44] on cell proliferation, migration, invasiveness and epithelial-mesenchymal transition (EMT) by analyzing BrdU incorporation, cell motility, invasiveness and EMT markers. Knockdown of TrkA by siRNA abolishes the NGF-induced effect on mitogenesis and invasiveness of CRPC cell lines, pointing to the specific role of TrkA in the observed responses. Analysis of the pathways implied in NGF signaling is also reported, pointing to the key role of MAPK as well as Akt activation in the observed effects. Finally, analysis in 3D cell culture further corroborates the finding that inhibition of TrkA impairs the NGF-elicited growth of organoids derived from CRPC cells.

## 2. Results

### 2.1. The Mitogenic Effect of NGF in CRPC Cells

Expression of TrkA as well as SRs in CRPC-derived C4-2B, DU145 and PC3 cells were analyzed by the western blot technique (Figure 1A). The anti-TrkA antibody revealed a 140 KDa band in lysates from the cell lines used. Of note, the amount of TrkA expressed in DU145 was higher than that observed in C4-2B and PC3 cells. Only C4-2B cells express AR. ERα was undetectable in all CRPC cell lines, while it was revealed in protein lysates from breast cancer MCF-7 cells, used as positive control. All the CRPC cell line used express ERβ, albeit to different extent. Protein lysates were also analyzed for E-cadherin and vimentin expression, as epithelial or mesenchymal markers, respectively. C4-2B, DU145 and PC3 cells express high levels of E-cadherin and low levels of vimentin (Figure 1A), although at varying levels indicating the epithelial signature of cell lines here employed. 

To evaluate the mitogenic effect of NGF, BrdU incorporation and 3-(4,5-dimethylthiazol-2-yl)-2,5-diphenyltetrazolium bromide (MTT) assays were done in CRPC-derived cells. Exposure of C4-2B (Figure 1B), DU145 (Figure 1C) and PC3 (Figure 1D) cells to NGF resulted in a significant increase in BrdU incorporation. The stimulatory effect induced by NGF is comparable to that elicited by serum stimulation of all the CRPC cell lines, suggesting that growth factors present in serum [45] significantly contributes to cell proliferation. TrkA inhibitor, GW441756 impairs the BrdU incorporation in NGF-challenged PC cells, indicating that TrkA activity is required for this effect. GW441756 does not significantly modify the BrdU incorporation of cell lines, when used alone, as control (Figure 1B–D) or in serum-stimulated cells (see the legend of Figure 1).

To reinforce the data obtained by BrdU incorporation, we also monitored cell proliferation by MTT assay. Consistent with findings obtained by BrdU analysis, MTT assay reveals that NGF treatment substantially stimulates the proliferation of all CRPC cell lines. Such stimulation started after 24h to reach the maximal effect after 72h NGF-treatment (Figure 1E–G). GW441756, which inhibits TrkA activity, does not affect the serum-induced proliferation, indicating its specific effect on TrkA signaling (Figure 1E–G). The finding that GW441756 significantly impairs the NFG mitogenic effect, without interfering in serum-elicited responses indicates that other growth factors (insulin-like growth factor, IGF), Platelet-derived growth factor (PDGF) [45]) are involved in serum-elicited response. 

Altogether, data in Figure 1 show that TrkA activation by NGF drives the DNA synthesis and proliferation in C4-2B (Figure 1B,E), DU145 (Figure 1C,F) and PC3 (Figure 1D,G) cells. 

### 2.2. NGF Promotes Migration and Invasiveness of CRPC Cells Through TrkA Activation

We next assessed whether NGF triggers the motility of CRPC cells. Therefore, a wound scratch assay was performed first. Quiescent C4-2B (panel A in Figure 2), DU145 (panel A in Figure 3) and PC3 (panel A in Figure 4) cells were wounded and then stimulated with NGF, in the absence or presence of GW441756. 

Cells were allowed to migrate and phase-contrast images were acquired. They show that the wound gap is significantly reduced in C4-2B (Figure 2A), DU145 (Figure 3A) and PC3 (Figure 4A) cells challenged with NGF. GW441756 inhibits the NGF-induced effect. 

Images captured at time 0 or from untreated cells are also shown for comparison. Results obtained from three different experiments are graphically presented in panel B of Figure 2, Figure 3 and Figure 4 for C4-2B, DU145 and PC3 cells, respectively. In the absence of any effect on cell proliferation (not shown), the wound gap is significantly (*p* < 0.05) reduced in cells treated with NGF, as compared with control, untreated cells. Here again, GW441756 reverses the effect of NGF in wound scratch assay.

We then analyzed the NGF effect on migration and invasiveness of CRPC cells using two additional assays. In collagen-coated Boyden’s chamber transmigration assay, NGF increases by ~2-, 1.9- and 3.2-fold the number of C4-2B (Figure 2C), DU145 (Figure 3C) and PC3 (Figure 4C) migrating cells, respectively. Here again, GW441756 significantly (*p* < 0.05) inhibits the NGF-induced effect, while it leaves almost unaffected the number of migrating cells, when used alone. Consistent with these results, significant changes in cell invasiveness were observed in Matrigel-coated Boyden’s chamber assay. Upon NGF treatment, indeed, the number of invading CRPC cells increases by ~2-fold in C4-2B (Figure 2D) and DU145 (Figure 3D) cells. A more robust effect by NGF (2.5-fold increase) is observed in PC3 cells (Figure 4D), likely because integrins, which mediate cellular adhesion to extracellular matrix (ECM), are highly expressed in PC3 cells, as compared with C4-2B and DU145 cells. Addition of GW441756 significantly (*p* < 0.05) inhibits the NGF-induced invasiveness, while it scantly affects the number of invading cells when used alone, as a control. As in proliferative assays (Figure 1), serum stimulation of C4-2B, DU145 and PC3 cells significantly increases the number of migrating or invading cells, and GW441756 does not affect these responses (Appendix A). Altogether, these findings indicate that TrkA activation by NGF promotes migration and invasiveness of CRPC cells in three different assays. 

### 2.3. NGF Promotes EMT of DU145 and PC3 cells Through TrkA Activation

EMT positively affects tumor progression and metastatic spreading by enabling a switch from the stationary epithelial-like cell phenotype to the motile mesenchymal phenotype, with high ability to migrate, invade, and disorganize the extracellular matrix (ECM; [46] and references therein). At the molecular level, EMT is characterized by a decrease in the expression of epithelial markers, mainly E-cadherin, which is located to the cell surface of epithelial tissues, and an increase in the expression of mesenchymal markers, such as vimentin, a cytoskeleton protein associated with EMT initiation [47].

To investigate whether NGF stimulation triggers EMT in CRPC cells, quiescent cells were left untreated or treated for 72 h with the indicated stimuli and protein lysates were analyzed for expression of E-cadherin and vimentin. Regardless of stimuli, the western blot analysis in Figure 5A (left panels) does not show significant changes in E-cadherin and vimentin levels in C4-2B cells. Similar findings were detected by densitometry analysis from three independent experiments (Appendix A). 

Under the same experimental conditions, NGF treatment does not induce significant morphological changes in C4-2B cells, and similar findings were observed in cells treated with GW441756, alone or in combination with NGF (right panels in Figure 5A). Quiescent DU145 cells (Figure 5B) and PC3 (Figure 5C) cells were then used. Here again, the cells were left untreated or treated for 72 h with the indicated compounds and lysates were analyzed for expression of E-cadherin and vimentin. NGF stimulation significantly down-regulates E-cadherin and up-regulates vimentin in both DU145 (Figure 5B) and PC3 cells (Figure 5C). Quantification of blots from three different experiments confirms these data (Appendix A, for DU145 cells; Appendix A for PC3 cells). GW441756 inhibits the NGF-elicited effect in both DU145 and PC3 cells, while it does not significantly modify E-cadherin and vimentin levels when used alone, as a control in both cell lines (Figure 5B,C; Appendix A).

We also looked at NGF-induced morphological changes of DU145 and PC3 cells. Images captured by phase-contrast microscopy show that 72h of NGF stimulation induces a rearrangement in DU145 (right panel in Figure 5B) and PC3 (right panel in Figure 5C) cell shape from an orthogonal epithelial cell morphology (untreated, right panels in Figure 5B,C), to a spindle-shaped fibroblast-like morphology (NGF-treated, right panels in Figure 5B,C), reminiscent of cells having undergone EMT. GW441756 inhibits the NGF-induced effect in both cell lines, while it is ineffective when used alone, as a control. 

Data in Figure 5 show that NGF robustly induces EMT in AR-negative DU145 and PC3 cells. In contrast, the NGF effect is almost undetectable in AR-positive C4-2B cells. This behavior is consistent with the finding that C4-2B cells move less than DU145 and PC3 cells upon NGF stimulation (Figure 2, Figure 3 and Figure 4).

Since C4-2B cells express AR, while DU145 and PC3 cells are both AR-negative, we hypothesized that the receptor negatively affects the aggressive phenotype of C4-2B cells in response to NGF. Therefore, we silenced AR by siRNA and analyzed the NGF effect on EMT, migration, invasiveness and proliferation of C4-2B cells. Cells were transfected in parallel with control siRNA. Irrespective of experimental condition, AR was actually silenced as shown by the Western blot in Figure 5D. Silencing of AR did not enhance the effect of NGF in EMT (Figure 5D), nor in migration assays (Figure 5E), nor in invasiveness assays (Figure 5G), as compared with control cells. Further, AR silencing did not significantly modify the BrdU incorporation in C4-2B cells challenged with NGF (Figure 5G). These findings indicate that NGF action in C4-2B cells is not related to AR, but additional characteristics that are intrinsic to these cells. 

Since the neurotrophin receptor p75NTR is considered a tumor and metastasis suppressor and its expression inversely correlates with PC progression [48], we also argued that in contrast with DU145 and PC3 cells, C4-2B cells still express the neurotrophin receptor p75NTR. The Western blot in Figure 6A shows that C4-2B cells express appreciable amounts of the p75NTR, which is undetectable in DU145 or PC3 cells. Thus, the absence of p75NTR is very likely responsible for the observed differences in the CRPC cell responsiveness to NGF. Nevertheless, the Western blot in Figure 6B and the densitometry analysis presented in the graphs in Figure 6C–E also reveals that DU145 and PC3 cells express high levels of proteins involved in the molecular machinery leading to cell locomotion, such as filamin A (Figure 6B,C), focal adhesion kinase (FAK; Figure 6B,D) and paxillin (Figure 6B,E), as compared with C4-2B cells.

In sum, our data, together with the observation that C4-2B cells express high levels of E-cadherin and have high adherent junction functionality [49], indicate that the absence of EMT as well as the finding that C4-2B move more slowly than DU145 and PC3 cells on NGF stimulation is very likely due to expression of p75(NTR) in C4-2B cells. The absence of p75(NTR) together with enrichment in the expression of genes that make up components and pathways associated with cell motility might allow a more aggressive phenotype in DU145 and PC3 cells.

### 2.4. NGF Increases CRPC 3D-Organoid Growth Through TrkA Activation

Interactions between tumor cells and ECM are critical for cancer progression [50]. To develop cell culture models resuming cancer tissues and to reproduce more faithfully the complex architecture of tumors in vivo, 3D cultures were performed. Phase-contrast images in Figure 7 show that a 3D structure was observed in all CRPC cell lines used on 4 days of culture in ECM. Consistent with previous findings [51], C4-2B cells generated larger, less differentiated and irregular organoids, as compared with DU145 and PC3 cells. DU145 and PC3 cells generated, instead, a roundish and well-differentiated organoids. At that day-culture, organoids were untreated or treated with NGF, in the absence or presence of GW441756. Changes in dimension and structure of organoids were monitored for 14 days and phase-contrast microscopy images were captured and shown (Figure 7). Quantification of data was also done and graphically presented (Appendix A). After 14 days, NGF increases by about 4-, 7- and 9-fold the size of C4-2B (upper panel in Figure 7, and Appendix A), DU145 (middle panel in Figure 7 and Appendix A) and PC3 (lower panel in Figure 7 and Appendix A) organoids, respectively. GW441756 significantly (*p* < 0.05 in Appendix A) inhibits the NGF-induced effect. Again, images in Figure 7 indicate that C4-2B cells form large, poorly differentiated organoids on 14 days of NGF treatment. In contrast, DU145 cells undergo a transition from differentiated, round DU145 organoids (basal condition) towards an invasive morphology when treated with NGF. Lastly, PC3 cells form round, well differentiated, and polarized spheroids, regardless of NGF stimulation. 

In addition to providing a valuable source for drug screening and more physiological information concerning the GW441756 inhibitory effect, findings in Figure 7 and Appendix A demonstrate for the first time a key role for TrkA activation in the growth of CRPC-derived organoids, which recapitulate tumor biology in vitro.

### 2.5. NGF Triggered TrkA-Tyr-490 Phosphorylation Controls MAPKs and Akt Activation in CRPC Cells

The binding of NGF to TrkA activates its kinase domain, thereby triggering various downstream pathways, such as the MAPK [52,53] and the phosphoinositide 3-kinases (PI3-K)/Akt (or PKB; protein kinase B) signaling cascade [31,54,55]. As readouts of NGF treatment, we analyzed MAPKs (Extracellular Signal-regulated Kinases (ERK1 and 2)) and Akt activation in a time course experiment. Quiescent CRPC cells were unstimulated or stimulated for the indicated times with NGF and protein lysates were analyzed by western blot technique. A robust ERK activation is observed in NGF-treated C4-2B cells, as compared with the weak Akt activation observed in the same experimental conditions. However, activation of both the effectors reached the maximal level within 5 minutes to decline within 15 and 30 min of NGF stimulation (Figure 8A). 

Again, NGF stimulation induces a striking ERK activation and a slight Akt activation in DU145 cells. ERK activation was observed already after 5 minutes of stimulation, with a stronger peak detectable after 30 minutes of treatment. Notably, Akt activation, if any, is very weak and delayed over the time, as compared with ERK activation. A slight P-Akt increase is detected only on 30 min NGF stimulation (Figure 8B). Finally, NGF stimulation induces ERK phosphorylation in PC3 cells, with an evident peak after 15 minutes of treatment. Remarkably, Akt activation is hardly detectable in NGF-treated cells (Figure 8C).

Densitometry analysis of immune-reactive bands detected by western blot shows that the NGF-induced increase in ERK phosphorylation over the basal was about 3-, 5- and 2.5-fold in C4-2B (Appendix A), DU145 (Appendix A) and PC3 (Appendix A) cells, respectively. Furthermore, the NGF-induced increase in Akt phosphorylation over the basal level was about 1.8- and 1.5-fold in C4-2B (Appendix A) and DU145 (Appendix A) cells, respectively. NGF did not modify the Akt activation status in PC3 cells (Appendix A), likely because of the absence of Phosphatase and Tensin Homolog (PTEN) in PC3 cells (Figure 8D). PTEN is, instead, robustly expressed in MCF7 cells, analyzed as positive control, and it is also expressed in C4-2B and DU145 cells (Figure 8D). However, the findings we observe are consistent with the concept that PTEN has a profound effect on NGF/TrkA signaling, since PTEN inhibits the NGF-mediated activation of the members of PI3-K/Akt signaling pathway, which is crucial for biological effects elicited by NGF in target cells [56]. 

Thus, albeit to a different extent and with different timing, NGF induces a strong ERK activation in all the CRPC cell lines, accompanied by a weak Akt activation in C4-2B and DU145 cells. We next investigated whether TrkA is the upstream event leading to NGF-induced ERK and Akt activation in CRPC cells. In addition to the tyrosine residues within the activation loop (Tyr-670, Tyr-674, and Tyr-675), TrkA exhibits two main tyrosine residues (Tyr-490 and Tyr-785), both outside the kinase domain. Once phosphorylated, these residues provide docking sites that lead to recruitment and activation of several signaling pathways [57,58]. Since MAPK and Akt pathway activation have been associated with Tyr-490 phosphorylation [59], we analyzed the NGF effect on Tyr490 phosphorylation of TrkA. Quiescent CRPC cells were left unchallenged or challenged for the indicated times with NGF, and P-Tyr490-TrkA phosphorylation was analyzed by Western blot technique. NGF treatment significantly increases P-Tyr-490 phosphorylation of TrkA in C4-2B (Figure 8E), DU145 (Figure 8F) and PC3 (Figure 8G) cells, respectively. GW441756 inhibits the NGF-induced effect, while it leaves unaffected P-Tyr490 TrkA when used alone, as a control (Figure 8E–G). Altogether, findings in Figure 8 and Appendix A suggest that Tyr-490 phosphorylation of TrkA represents the initial event activated by NGF in CRPC cells. 

NGF-induced tyrosine phosphorylation of the TrkA receptor controls the subsequent activation of ERK or Akt [60]. Quiescent C4-2B (Figure 8H), DU145 (Figure 8I) and PC3 (Figure 8L) cells were left unchallenged or challenged for the indicated times with NGF, in absence or presence of GW441756. The western blots in Figure 8 show that NGF treatment significantly increases ERK activation. GW441756 inhibits such effect in all CRPC cells used, while it leaves almost unaffected ERK activation when used alone, as control (Figure 8H–L). Again, NGF treatment increases Akt activation in C4-2B (Figure 8H) and in DU145 (Figure 8I) cells. Here again, GW441756 inhibits the NGF-induced effect, while it does not affect Akt activation when used alone, as control (Figure 8H–L). Consistent with data in panel C, neither NGF, nor GW441756 modify Akt activation (Figure 8L).

By interrogating the Search Tool for the Retrieval of Interacting Genes/Proteins (String) database, which contains the known or predicted protein-protein interactions [61], we then constructed a network diagram for both TrkA (NTRK1) and ERK (MAPK1; Figure 9A) as well as TrkA (NTRK1) and Akt (AKT1; Figure 9B), respectively. Consistent with findings presented in Figure 8, these proteins belong to the same network through known or predicted interactions.

### 2.6. The Role of TrkA in NGF-Induced Proliferation and Invasiveness of CRPC Cells

Finally, we investigated the role of TrkA in the NGF-elicited effects by siRNA approach. To this end, C4-2B, DU145 and PC3 cells were transfected with TrkA siRNA and the NGF effect on BrdU incorporation and invasiveness of cells was monitored. Cells were transfected in parallel with control siRNA. The Western blot in Figure 9 shows that, irrespective of experimental condition, TrkA was actually silenced in C4-2B (C), DU145 (D) and PC3 (E) cells. Silencing of TrkA abolished the NGF-induced increase in BrdU incorporation of C4-2B (Figure 9F), DU145 (Figure 9G) and PC3 (Figure 9H), as compared with control cells. Again, knockdown of TrkA drastically impaired the NGF effect on invasiveness of C4-2B (Figure 9I), DU145 (Figure 9L) and PC3 (Figure 9M), as compared with control cells. Taken together, these findings indicate that in our experimental setting, the NGF action is strictly related to TrkA.

### 2.7. The Role of MAPK and Akt Activation in NGF-Induced Proliferation and Migration of CRPC Cells 

The role of NGF-induced activation of MAPK and Akt on CRPC cells proliferation and migration was then investigated using the PI-3K inhibitor, LY-294002 [62,63] as well as the Mitogen-activated protein kinase kinase (MEKK) inhibitor, PD98059 [31]. As expected, exposure of C4-2B (Figure 10A), DU145 (Figure 10B) and PC3 (Figure 10C) cells to NGF increases BrdU incorporation. LY-294002 prevents the NGF-elicited S-phase entry of C4-2B (Figure 10A) and DU145 (Figure 10B), while it does not modify the NGF-induced BrdU incorporation in PC3 cells (Figure 10C). These results are consistent with the absence of PTEN in PC3 cells. LY-294002 doesn’t significantly affect BrdU incorporation of CRPC cells, when used alone. Albeit to a different extent, PD98059 prevents the BrdU incorporation of C4-2B (Figure 10A), DU145 (Figure 10B) and PC3 (Figure 10C) cells stimulated with NGF. The inhibitor doesn’t significantly affect BrdU incorporation of DU145 and PC3 cells, while it significantly increases the proliferation of C4-2B cells when used alone, likely because MEK inactivation paradoxically leads to the simultaneous hyper-activation of PI3-K/Akt signaling in C4-2B cells. Consistent with this hypothesis, it has been previously reported that treatment with MAPK inhibitor alone induces the compensatory activation of phosphorylated/activated Akt in hepatocarcinoma cells [64]. 

To evaluate the effect of LY-294002 and PD98059 on NGF-induced CRPC cell migration, collagen-coated Boyden’s chambers were used. Both the inhibitors significantly impair the NGF-induced effect, while they leave almost unaffected the number of migrating cells, when used alone Figure 10D–E). 

Altogether, findings in Figure 10 indicate that both Akt and MAPK activities are required for the NGF-induced S-phase entry and migration of CRPC cells. Noticeably, while the NGF migratory effect in CRPC cells is more sensitive to MEK inhibition, the NGF proliferative action is more sensitive to PI3-K inhibition, with the exception of PC3 cells. Thus, NGF utilizes PI3-K pathway to transmit proliferative and survival signaling, while it engages MAPK signaling to activate the machinery involved in motility of CRPC cells. 

## 3. Discussion

PC progression is accompanied by modifications in the expression of NGF and neurotrophin receptors [65,66]. These changes mainly consist in a reduction of p75NTR expression by the epithelial cells, in the absence of any significant modification in the expression of NGF or TrkA [67]. Immunohistochemistry (IHC) studies in normal and malignant prostate epithelial tissue have substantially confirmed these findings [68]. PC cells, however, acquire the ability to express neurotrophins as they progress and prostatic stromal cells from PC patients release in tumor microenvironment appreciable levels (from 827 to 2027 pg/10^6^/48h) of neurotrophins [69,70]. All these findings point to the role of neurotrophin/Trk axis in sustaining PC progression and spreading.

In this paper, we have investigated the effect of NGF in various CRPC-derived cells. Among them, C4-2B cells express considerable amounts of AR, while DU-145 and PC3 are AR-negative. Nevertheless, CRPC cell lines often harbor AR variants that lack the C-terminal ligand-binding domain. As such, these variants are frequently constitutively active, since they do not require the presence of androgens to initiate downstream AR signaling and their expression has been correlated with androgen deprivation therapy (ADT) resistance [71,72]. Thus, it cannot be excluded that the CRPC cell lines here employed express AR variants that might differentially control the NGF signaling. The cell lines here used, however, harbor significant levels of TrkA as well as ERβ, while they lack ERα. Additionally, they release undetectable levels of NGF in cell medium [24]. As such, their treatment with 100 ng/mL NGF, which is considered the optimal NGF concentration [24,25,31], robustly increases Tyr-490 phosphorylation of TrkA. This event is upstream of NGF-induced ERK and/or Akt activation in CRPC cells. A significant increase of BrdU incorporation and cell proliferation then follows, as GW441756 similarly inhibits Tyr-490 phosphorylation of TrkA and cell proliferation. The proliferative effect of NGF is similar in all cell lines employed and is reverted by TrkA inhibitor, GW441756 as well as by knockdown of TrkA. In addition to indicating that TrkA activation is required for MAPK as well as Akt activation and the consequent progression in cell cycle, our findings suggest that NGF can substitute for androgens in sustaining CRPC cell survival. 

We also analyzed the NGF effect on cell motility and invasiveness by wound scratch and Trans-migration assays. Our data show that NGF induces cell motility, spreading as well as EMT in DU-145 and PC3 cells. These results are consistent with the concept that cells undergoing EMT become more motile [47]. GW441756 and TrkA siRNA both abolished the NGF effect on CRPC cell invasiveness, further corroborating the valuable effect of TrkA inhibitors in impairing the CRPC spreading. Albeit at lower extent, C4-2B cells still migrate upon NGF stimulation, in the absence of EMT. These results are only apparently conflicting, since acquisition of a mesenchymal cell state is not a prerequisite of a migratory phenotype in vitro and in vivo [73]. Further, C4-2B cells express high levels of E-cadherin and have high adherent junction functionality. As such, they cannot fully respond to extracellular EMT inducers [49]. Again, in contrast with DU145 and PC3, C4-2B cells harbor appreciable levels of p75 (NTR), whose expression mitigates PC spreading [48]. Finally, the possibility that NGF does not fully activate the signaling network required for C4-2B cell locomotion (e.g., small GTPases) cannot be excluded. We have consistently observed that DU145 and PC3, but not C4-2B cells are enriched of signaling effectors required for cell locomotion and invasiveness, such as filamin A, focal adhesion kinase and paxillin [74,75,76]. Moreover, the possibility that the full-length AR expressed in C4-2B negatively controls the NGF action is excluded by the finding that AR knockdown does not significantly modify the NGF effect on migration, invasiveness, EMT and mitogenesis of C4-2B cells (Figure 5). Thus, in contrast with our previous results in LNCaP cells [32], NGF action does not seem related to AR/TrkA crosstalk in C4-2B cells. These findings call for comments. LNCaP and C4-2B cells exhibit important differences in the number and distribution of mutations as well as gene expression. Among them, important changes in the focal adhesion and ECM-receptor interaction pathways have been reported in C4-2B cells [77]. These differences might be crucial in dictating the choice of AR partners. Thus, in LNCaP cells, AR might easily associate with growth factor receptors (e.g., TrkA or EGF-R, [32,78]), while in C4-2B cells, the receptor might recruit other signaling effectors (e.g., FAK and/or proteins involved in integrin/adhesion pathways). In favor of this hypothesis, we have reported that various factors, including cell type and ligand concentration, regulate the identity of partners associated with AR [33]. Other mechanisms, however, might explain the observed difference. C4-2B cells, for instance, express AR8 variant that controls the EGF-induced Src activation and the consequent AR phosphorylation [79]. Expression of this variant might negatively impact the responsiveness to NGF as well as the AR/TrkA crosstalk in C4-2B cells. Again, it has been recently shown that the ubiquitin ligase TRAF4 promotes TrkA ubiquitination and hyperactivates TrkA kinase activity, thereby leading to PC metastasis [80]. Therefore, it cannot be excluded the possibility that various TRAF4 levels differentially modulate the AR/TrkA crosstalk in C4-2B cells as well as the strength and duration of NGF-induced TrkA activation among the various CRPC cells. Further investigations would definitely clarify this issue. 

Organoid culture is largely accepted to study stem cells, organ development and patient-specific diseases [81]. A standard protocol for organoid culture of human and mouse prostate epithelial and PC tissues has been previously described [82]. However, the NGF effect on organoid growth of the commonly used CRPC cell lines has not been reported so far. In our study, we followed the reported protocol [82] to set up organoid culture of CRPC cells. We found that C4-2B, PC3 and DU-145 organoids were formed after 14-days culture in hydrogel and that NGF significantly enhanced the growth of CRPC cells in 3D. Here again, GW441756 reverted the NGF effect further suggesting that specific targeting of NGF/TrkA axis represents a very promising tool in CRPC therapy. 

Strategies aimed at inhibiting NGF signaling are already in clinical trials for the treatment of PC. On the basis of preclinical findings, the pan-Trk inhibitor CEP-701 entered clinical trials in PC patients and encouraging data were obtained by initial phase I studies, although the antitumor effects observed with CEP-701 may be caused by its ability to inhibit multiple kinases that are needed for survival and proliferation of PC cells [83]. Moreover, CEP-701 is not a pure Trk inhibitor, since it can also inhibit vascular endothelial growth factor receptor 2 (VEGFR2), Platelet-derived growth factor receptor, beta (PDGFRβ and Rearranged during Transfection gene (RET) [84]. These findings, together with the observation that Receptor Tyrosine Kinase (RTK) inhibitors frequently induce side effects and resistance ([29,35], and therein refs) indicate that only a precise understanding of the molecular landscape in human PC specimens might allow a more tailored therapy. Therefore, 3D model from biopsies of PC patients can be envisaged for identification of neurotrophins/Trk signalling derangements as well as discovery of new drugs specifically targeting this circuit in aggressive PC. Thus, 3D models that more closely reflect the setting of PC might improve translational research and preclinical drug screening. Finally, the finding that NGF/TrkA signaling also controls proliferation and invasiveness of AR-positive C4-2B cells strongly suggests that targeting of this axis might be of value in the clinical approach of ADT-refractory PC. This strategy would also overcome ADT resistance often observed in PC patients exhibiting AR variants, thereby improving their clinical outcome and disease-free survival.

## 4. Materials and Methods

### 4.1. Chemicals and Reagents

NGF (Millipore; Burlington, MA, USA) was used at 100 ng/mL. The TrkA inhibitor, GW441756 (Selleckchem, Munich, Germany) was added (at 1 μM, final concentration) 20 minutes before NGF stimulation. The mitogen-activated kinase kinase (MEKK) inhibitor PD98059 (Alexis, San Diego, CA, USA) was added (at 10 μM, final concentration) 20 minutes before NGF stimulation. The PI-3K inhibitor, LY-294002 (Millipore; Burlington, MA, USA) was added (at 10 μM, final concentration) 30 minutes before NGF stimulation [85].

### 4.2. Cell Cultures

Human PC-derived DU145 cells were from Cell Bank Interlab Cell Line Collection (ICLC, Genova, Italy). Human PC-derived PC3 cells were a gift of Dr. P. Limonta (Department of Pharmacological and Biomolecular Sciences, Università degli Studi di Milano, Milano, Italy). Human PC-derived C4-2B cells were a gift of Dr G.N. Thalmann (Department of Urology, University Hospital Bern, Bern, Switzerland). Cells were maintained at 37 °C in humidified 5% CO_2_ atmosphere. DU145 cells were cultured in phenol-red DMEM containing 10% fetal bovine serum (FBS), penicillin (100 U/mL), streptomycin (100 U/mL) and glutamine (2 mM). Twenty-four hours before stimulation, growing DU145 cells at 70% confluence were made quiescent using phenol red-free Dulbecco’s Modified Eagle Medium (DMEM) medium containing penicillin (100 U/mL) and streptomycin (100 U/mL). PC3 cells were cultured in phenol-red Roswell Park Memorial Institute (RPMI)-1640/F12 containing 10% fetal bovine serum (FBS), penicillin (100 U/mL), streptomycin (100 U/mL) and glutamine (2 mM). Forty-eight hours before stimulation, growing cells at 70% confluence were made quiescent using phenol red-free RPMI-1640 medium supplemented with penicillin (100 U/mL), streptomycin (100 U/mL), and 0,5% charcoal-stripped serum (CSS). C4-2B cells were cultured in RPMI-1640 supplemented with 10% FBS, glutamine (2 mM), penicillin (100 U/mL), streptomycin (100 U/mL), sodium pyruvate (1 mM) and non-essential amino acids (10mM). Seventy-two hours before stimulation, growing cells at 70% confluence were made quiescent using phenol red-free RPMI-1640 medium containing 10% CSS, penicillin (100 U/mL), streptomycin (100 U/mL) and glutamine (2 mM). Media and supplements were from Gibco (Thermofisher; Waltham, MA USA). Human BC-derived MCF-7 cells were from Cell Bank Interlab Cell Line Collection (ICLC). Cells were maintained at 37 °C in humidified 5% CO_2_ atmosphere and cultured in phenol-red DMEM containing 10% fetal bovine serum (FBS), penicillin (100 U/mL), streptomycin (100 U/mL) and glutamine (2 mM). The cell lines employed were routinely monitored for *Mycoplasma* contamination and expression of sex steroid receptors, as previously reported [86].

### 4.3. Transfection and siRNA experiments

siRNA experiments were done as described (Castoria et al., 2014). Briefly, for AR siRNA was employed a pool of four target-specific 19–25nt siRNAs (sc-29204; Santa Cruz; Dallas, TX, USA). Non-targeting siRNAs, containing a scrambled sequence, was from Santa Cruz. siRNA was transfected using Lipofectamine^TM^ 2000 (Thermofisher). For TrkA siRNA was employed a pool of 3 target-specific 19–25 nt siRNAs designed to knock down gene expression (sc-36726; Santa Cruz, Dallas, Texas, USA). When indicated, cells were co-transfected with siRNA Alexa Fluor 488 (Cell Signalling, Danver, MA, USA) to identify transfected cells. Six hours after transfection, cells were made quiescent, then used.

### 4.4. DNA Synthesis, Immunofluorescence (IF) Microscopy and MTT Assay

Quiescent cells were left unchallenged or challenged with NGF (100 ng/mL), in the absence or presence of the indicated compounds for 18h. After in vivo pulse with BrdU (100 µM final concentration; Sigma-Aldrich, St. Louis, MO, USA), the BrdU incorporation into newly synthesized DNA was analyzed by IF microscopy, using a DMLB Leica (Leica, Wetzlar, Germany) fluorescent microscope equipped with HCX PL Apo 63× oil objective, as reported [87]. The BrdU incorporation was calculated by the formula: percentage of BrdU-positive cells = (No. of BrdU-positive cells/No. of total cells) × 100. MTT assay was done using WST-1 reagent (Roche, Basel, Switzerland), as described [88]. Values were expressed as fold increase over the basal level.

### 4.5. Miniaturized 3D Cultures in ECM

The embedding method has been used to establish organoids for drug treatment experiments [89]. Briefly, cell suspension containing 5 × 10^4^ cells was mixed with 200 μL of VitroGel-3D-RGD (The Well Biosciences, North Brunswick, NJ, USA) for each well. The mixture was pipetted in 24-well plate as reported [90] and allowed to solidify for 45 min at 37 °C, before the addition of 400 μL organoid plating medium to each well. Organoid plating medium was made as reported [82], using phenol red-free DMEM/F12 medium, containing 7% CSS, penicillin (100 U/mL), streptomycin (100 U/mL), diluted GlutaMAX 100X, 10 mM Hepes, B27 (50 × stock solution), 1M nicotinamide, 500 mM N-acetylcysteine and 10 μM Y-27632 (Millipore, Burlington, MA, USA). After 3 days, the organoid-plating medium was replaced with a similar medium without N-acetylcysteine and Y-27632. At the 4th day, organoids were untreated or treated with the indicated stimuli, in the absence or presence of inhibitors. The medium was changed every 3 days. Different fields were analyzed using DMIRB Leica (Leica) microscope equipped with C-Plan 20× objective (Leica). At the indicated times, phase-contrast microscopy images were acquired using a DFC 450C camera (Leica) and the Application Suite Software (Leica). Images are representative of three independent experiments. The relative organoid size (area) was calculated using the Application Suite Software and expressed as a fold increase over the organoid area calculated at 4th day. 

### 4.6. Wound Scratch Analysis, Migration, Invasiveness and Phase-contrast Microscopy

For wound scratch analysis, 1.8 × 10^5^ cells were seeded in a 24-well plate. Cells were made quiescent as above described and then wounded using 10µl sterile pipette tips. Cells were washed with PBS and then left un-stimulated or stimulated for the indicated times with NGF (100 ng/mL), in the absence or presence of the indicated compounds. To avoid cell proliferation, cytosine arabinoside (Sigma-Aldrich) at 50 µM (final concentration) was included in the cell medium. Different fields were analyzed using DMIRB inverted microscope (Leica) equipped with N-Plan 10× objective (Leica), as reported [91]. Phase-contrast images were captured using a DFC 450C camera (Leica) and acquired using Application Suite Software (Leica). Images are representative of at least three different experiments. The wound gap was calculated using Image J Software and expressed as % of the decrease in the wound area. Migration assay was done using 3 × 10^4^ cells in Boyden’s chambers with 8 μm polycarbonate membrane (Corning; Corning, NY, USA) pre-coated with collagen [92]. The indicated stimuli were added to the upper and the lower chambers. Cytosine arabinoside (at 50 µM) was included in the cell medium. After 7 h, non-migrating cells from the membrane upper surface were removed using a sterile cotton swab. Invasion assay was done using 5 × 10^4^ cells in Boyden’s chambers with 8 μm polycarbonate membrane (Falcon) pre-coated with growth factor reduced and phenol red-free Matrigel (Corning; Corning, NY, USA), as reported [92]. The indicated stimuli were added to the upper and the lower chambers. Here again, cytosine arabinoside (at 50 µM) was included in the cell medium. After 24 h, non-invading cells from the membrane upper surface were removed using a sterile cotton swab. In both, migration and invasiveness assays, the membranes were fixed for 20 min in 4% paraformaldehyde, stained with Hoechst, removed with forceps from the companion plate and mounted. Migrating or invading cells from at least 30 fields/each membrane were counted as described [92], using a DMLB (Leica) fluorescent microscope, equipped with HCPL Fluotar 20× objective. Data are representative of at least three different experiments.

### 4.7. EMT, Lysates and Western Blot Technique 

EMT markers were analyzed by western blot technique, as reported [93]. Briefly, quiescent cells were left un-stimulated or stimulated for the indicated times with NGF, in the absence or presence of GW441756 and then harvested in PBS containing 5 mM Ethylene Diamine Triacetic Acid (EDTA). Cell pellets were washed twice by centrifugation with PBS at 1,200 rpm and protein lysates were prepared [93]. Sodium Dodecyl Sulfate Polyacrylamide Gel Electrophoresis (SDS-PAGE) and Western blot techniques were done according to the same report. The rabbit polyclonal anti-vimentin (H-84; Santa Cruz Biotechnology) antibody was used to detect vimentin. The mouse monoclonal anti E-cadherin (clone 36/E, BD Bioscience; San Jose, CA) antibody was used to detect E-cadherin. AR was revealed as reported [86], using the mouse monoclonal anti-AR (441; Santa Cruz Biotechnology). The rabbit polyclonal anti-ERα (HC-20; Santa Cruz Biotechnology) and anti-ERβ (06-629, Millipore) antibodies were used to detect ER (α or β), respectively. The mouse monoclonal anti-tubulin antibody (Sigma-Aldrich) was used to detect tubulin. TrkA, ERK and Akt phosphorylation were analyzed as reported [31,32], using the rabbit polyclonal anti p-TrkA (Tyr-490; 9141S, Cell Signaling), or the mouse monoclonal anti p-ERK (sc-7383; Santa Cruz Biotechnology), or the rabbit polyclonal anti p-Akt (Ser-473; 9271S; Cell Signaling) antibodies. The rabbit polyclonal anti-TrkA (06-574; Millipore), or anti-ERK (C-14; Santa Cruz Biotechnology), or anti-Akt (9272; Cell Signaling) antibodies were used to detect TrkA, ERK and Akt, respectively. The mouse monoclonal antibody NGFR p75 (B-1; sc271708; Santa Cruz Biotechnology) antibody was used to detect p75NTR. Filamin A, FAK and Paxillin were detected using the rabbit polyclonal anti-Filamin A (9272; Cell Signaling), the mouse monoclonal anti-FAK (BD Bioscience) and the mouse monoclonal anti-paxillin antibodies (clone 349; BD Bioscience), respectively. The ECL system (GE Healthcare, Chicago, Illinois, USA) was used to reveal immuno-reactive proteins. 

### 4.8. Analysis of Network Construction

Network diagrams for protein-protein interaction analysis were generated with the use of Search Tool for the Retrieval of Interacting Genes software (STRING; https://string-db.org/). Interaction prediction was performed using the list of gene/protein names as a query. The results of this analysis yielded a gene/protein interaction network, with the intensity of the edges reflecting the strength of the interaction score, as reported in the legend. STRING database builds a protein-protein interaction network for all of the required proteins [94].

### 4.9. Statistical Analysis

All the experiments were performed in triplicate and data are presented as mean ± standard deviation. Comparison for the different assays was evaluated with the paired two-tailed Student’s *t*-test. We used a *p* value of ≤ 0, 05 indicative of statistical significance. 

## 5. Conclusions

TrkA is involved in various cellular activities, mainly in neuritogenic and trophic signals [17,18,19]. Deregulation of TrkA is also a common feature of several cancer types, including PC [20,23].

By activating TrkA and the downstream pathways (MAPK and Akt), NGF leads to proliferation and invasiveness in three different CRPC cell lines. Moreover, NGF/TrkA axis promotes the acquisition of a more aggressive phenotype in miniaturized 3D model. This approach offers significant advantages over other systems for the study of PC features. Furthermore, in the era of precision medicine, 3D model represents an invaluable tool to select compounds for a more tailored management of metastatic CRPC. Our findings warrant further prospective studies in 3D models from patients to validate the reliability of TrkA as a biomarker to track and target in metastatic CRPC. 

## Figures and Tables

**Figure 1 cancers-11-00784-f001:**
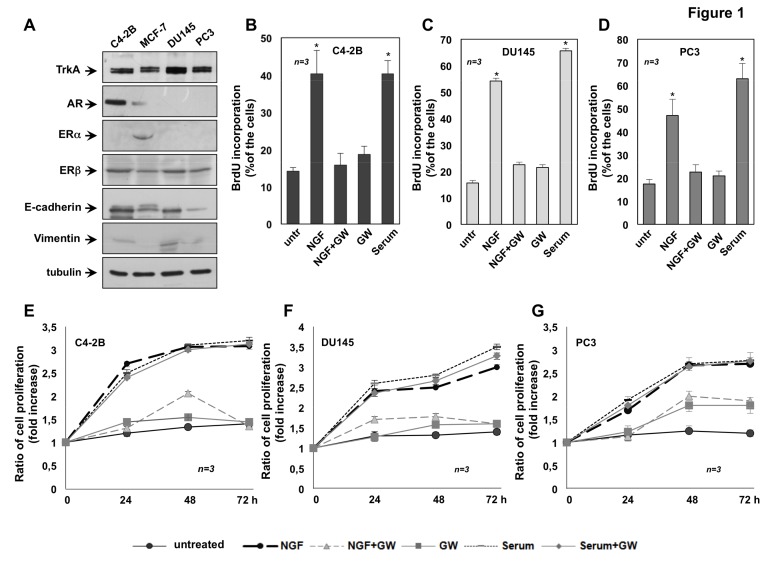
Tropomyosin receptor kinase A (TrkA) activation mediates the nerve growth factor (NGF) mitogenic effect in castration-resistant prostate cancer (CRPC) cells. C4-2B, DU145 and PC3 cells were used. In **A**, lysates from the indicated cell lines were prepared and proteins were analyzed by Western blot, using the antibodies against the indicated proteins. Quiescent C4-2B (**B**), DU145 (**C**) and PC3 (**D**) cells were left untreated (untr) or treated for 18h with the indicated compounds. After in vivo pulse with 100 μM BrdU, BrdU incorporation was analyzed by IF and expressed as % of total cells. Quiescent C4-2B, DU145 and PC3 cells were left untreated (untr) or treated for 18h with 20% (v/v) serum, in the absence or presence of GW441756. After in vivo pulse with 100 μM BrdU, BrdU incorporation was analyzed by IF and expressed as % of total cells. Under basal conditions, 9%, 12% and 14% of C4-2B, DU145 and PC3 cells, respectively, incorporated BrdU. Serum stimulation increased by 45%, 67% and 59% the number of BrdU incorporating C4-2B, DU145 and PC3 cells, respectively. Addition of GW441756 did not significantly inhibit this number (43%, 63% and 57% for C4-2B, DU145 and PC3 cells, respectively). Quiescent C4-2B (**E**), DU145 (**F**) and PC3 (**G**) were left untreated or treated for 24, 48 and 72 h with the indicated compounds. Cell proliferation was assayed using the WST-1 (water soluble tetrazolium-1) reagent. Graphs in **E**–**G** represent the ratio of proliferation, which was expressed as fold increase over the basal absorbance. NGF stimulation induced a significant variation of the proliferation as compared with untreated cells (*p* < 0.05). In **B**–**G**, NGF was used at 100 ng/mL; GW441756 (GW) was used at 1μM. When indicated, serum was used at 20% (v/v). Three independent experiments were done. Means and standard error of the means (SEMs) are shown. *n* represents the number of experiments. * *p* < 0.05 for the indicated experimental points vs. the corresponding untreated control.

**Figure 2 cancers-11-00784-f002:**
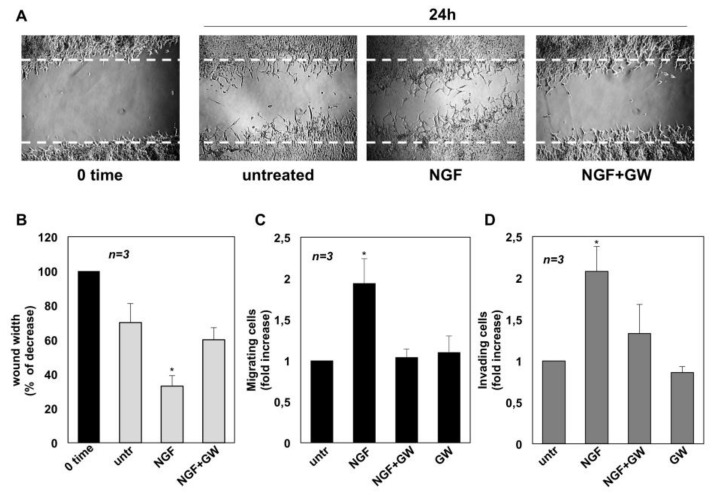
Nerve growth factor (NGF) triggers migration and invasiveness in C4-2B cells. In **A**, quiescent C4-2B cells were wounded and left untreated or treated with NGF for the indicated times. GW441756 (GW) was added at 1μM. Phase-contrast images are representative of three different experiments, each in duplicate. In (**B**), the wound area was measured using Leica Suite Software and data are presented as % in wound width over the control cells, analyzed at time 0. Means and standard error of the means (SEMs) are shown. *n* represents the number of experiments. Quiescent C4-2B cells were used for migration (**C**) and invasion (**D**) assays in Boyden’s chambers pre-coated with collagen or Matrigel, respectively. The indicated compounds were added to the upper and the lower chambers. NGF was used at 100 ng/mL and GW441756 (GW) at 1 μM. After 7 h (in **C**) or 24 h (in **D**), migrating or invading cells were counted as reported in Methods. Results from three different experiments were collected and expressed as fold increase. Means and SEMs are shown. *n* represents the number of experiments. * p < 0.05 for the indicated experimental points vs. the corresponding untreated control.

**Figure 3 cancers-11-00784-f003:**
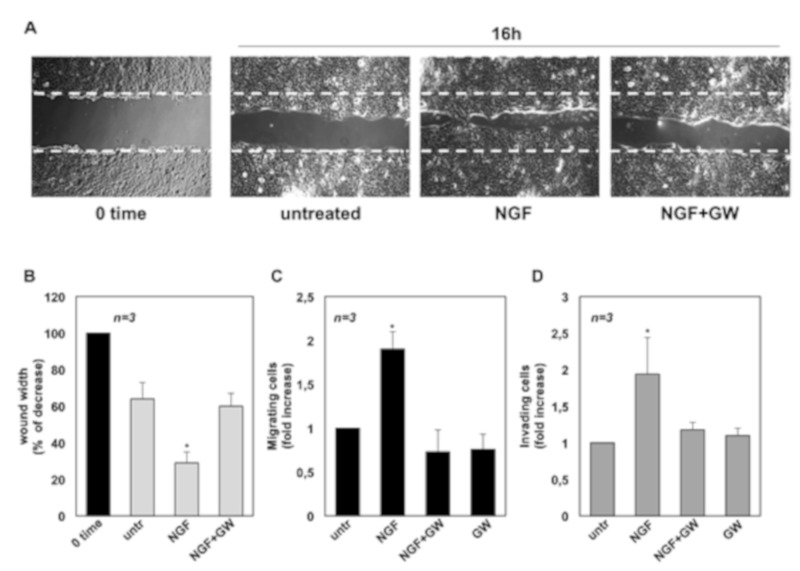
Nerve growth factor (NGF) triggers migration and invasiveness in DU145 cells. In **A**, quiescent DU145 cells were wounded and then left untreated or treated with NGF for the indicated time. When indicated, GW441756 (GW) was added at 1 μM. Phase-contrast images are representative of three different experiments, each in duplicate. In (**B**), the wound area was measured as in Figure 2 B and data are presented as % in wound width over the control cells, analyzed at time 0. Means and SEMs are shown. *n* represents the number of experiments. Quiescent C4-2B cells were used for migration (**C**) and invasion (**D**) assays in Boyden’s chambers pre-coated with collagen or Matrigel, respectively. The indicated compounds were added to the upper and the lower chambers. NGF was used at 100 ng/mL and GW441756 (GW) was used at 1 μM. After 7 h (**C**) or 24 h (**D**), cells were counted as reported in Methods. Results from three different experiments were collected and expressed as fold increase. Means and standard error of the means (SEMs) are shown. *n* represents the number of experiments. * p < 0.05 for the indicated experimental points vs. the corresponding untreated control.

**Figure 4 cancers-11-00784-f004:**
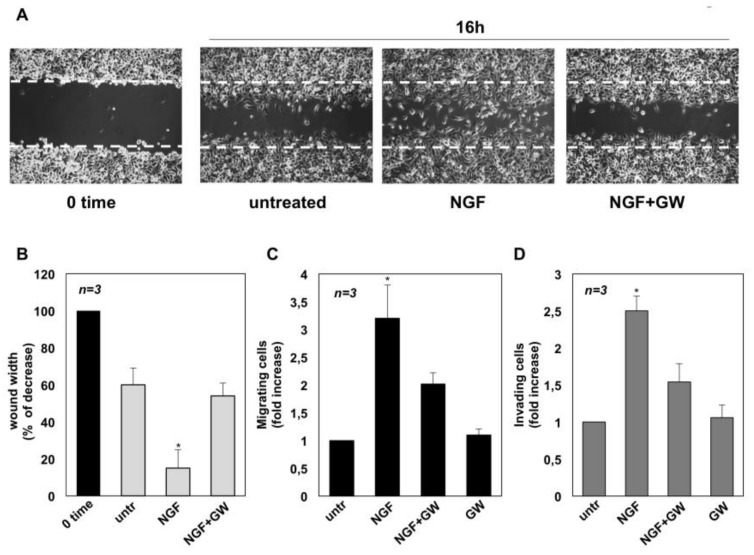
Nerve growth factor (NGF) triggers migration and invasiveness in PC3 cells. In **A**, quiescent PC3 cells were wounded and then left untreated or treated with NGF for the indicated times. GW441756 (GW) was added at 1μM. Phase-contrast images are representative of three different experiments, each in duplicate. In (**B**), the wound area was measured as in Figure 2B and data are presented as % in wound width over the control cells analyzed at time 0. Means and SEMs are shown. *n* represents the number of experiments. Quiescent C4-2B cells were used for migration (**C**) and invasion (**D**) assays in Boyden’s chambers pre-coated with collagen or Matrigel, respectively. The indicated compounds were added to the upper and the lower chambers. NGF was used at 100 ng/mL and GW441756 (GW) was used at 1μM. After 7 h (**C**) or 24 h (**D**), migrating or invading cells were counted as reported in Methods. Results from three different experiments were collected and expressed as fold increase. Means and standard error of the means (SEMs) are shown. *n* represents the number of experiments. * *p* < 0.05 for the indicated experimental points vs. the corresponding untreated control.

**Figure 5 cancers-11-00784-f005:**
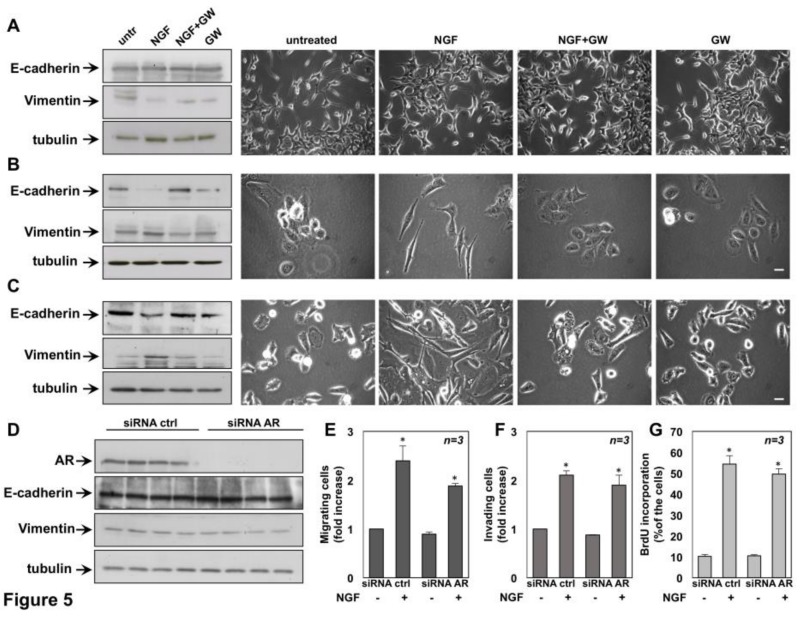
Nerve growth factor (NGF) promotes epithelial-mesenchymal transition (EMT) in DU145 and PC3 cells and silencing of Androgen Receptor (AR) does not affect EMT, migration, invasiveness and DNA synthesis of NGF-treated C4-2B cells. Quiescent C4-2B (**A**), DU145 (**B**) and PC3 (**C**) cells were left untreated or treated for 72h with the indicated compounds. NGF was used at 100ng/mL and GW441756 (GW) at 1 μM. In **A-C** (left panels), protein lysates were analyzed by Western blot, using the antibodies against the indicated proteins. The blots are representative of three different experiments. In **A-C** (right panels), Phase-contrast images are representative of 3 different experiments, each in duplicate. Scale bar, 10 μm in **A** and 90 μm in **B** and **C**. In **D-G,** C4-2B cells were transfected with AR siRNA or control siRNA, as reported in Methods. After transfection, the cells were made quiescent and then left un-stimulated or stimulated as reported. In **D**, cells were stimulated as in **A** and protein lysates were analyzed by western blot using the antibodies against the indicated proteins. Cells were used for migration (**E**) and invasion (**F**) assays in Boyden’s chambers pre-coated with collagen or Matrigel, respectively. The indicated compounds were added to the upper and the lower chambers. After 7 h (in **E**) or 24 h (in **F**), migrating or invading cells were counted as reported in Methods. In **G**, transfected C4-2B cells on coverslips were left untreated (untr) or treated for 18h with the indicated compounds. After in vivo pulse with 100 μM BrdU, BrdU incorporation was analyzed by immunofluorescence (IF) and expressed as % of total cells. In **E**–**G**, the cells were co-transfected with siRNA Alexa Fluor 488 to help identification of transfected cells. Results from three different experiments were collected and expressed as fold increase. Means and standard error of the means (SEMs) are shown. *n* represents the number of experiments. * *p* < 0.05 for the indicated experimental points vs. the corresponding untreated control.

**Figure 6 cancers-11-00784-f006:**
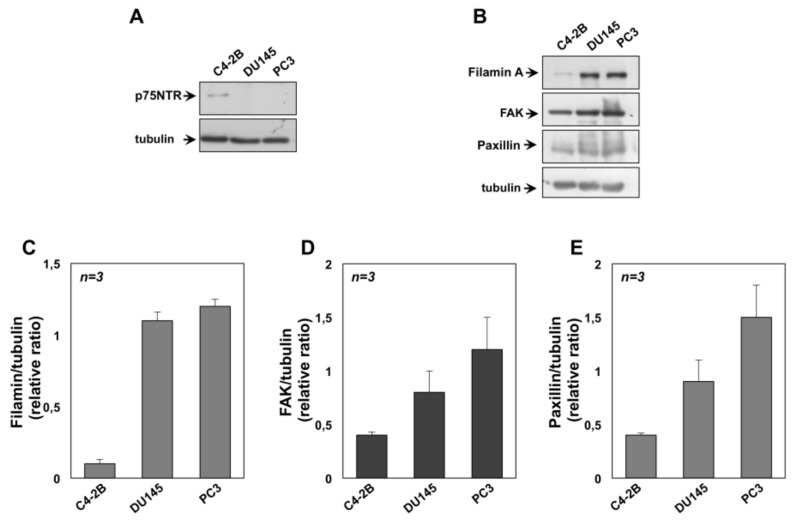
Expression of p75NTR and signaling effectors involved in locomotion of castration-resistant prostate cancer (CRPC) cells. C4-2B, DU145 and PC3 cells were used. In **A**, lysates from the indicated cell lines were prepared and proteins were analyzed by western blot, using the antibodies against p75NTR. In **B**, lysates from the indicated cell lines were prepared and proteins were analyzed by Western blot, using the antibodies against the indicated proteins. Filters were re-probed using anti tubulin antibody, as a loading control. Western blots in **A** and **B** are representative of three different experiments. Expression levels of proteins were analyzed by densitometry analysis, using NIH Image J Software. The ratio between Filamin A/tubulin (**C**), FAK/tubulin (**D**) and Paxillin/tubulin (**E**) was evaluated. Results were expressed as relative ratio. Means and SEMs are shown, *n* represents the number of the experiments.

**Figure 7 cancers-11-00784-f007:**
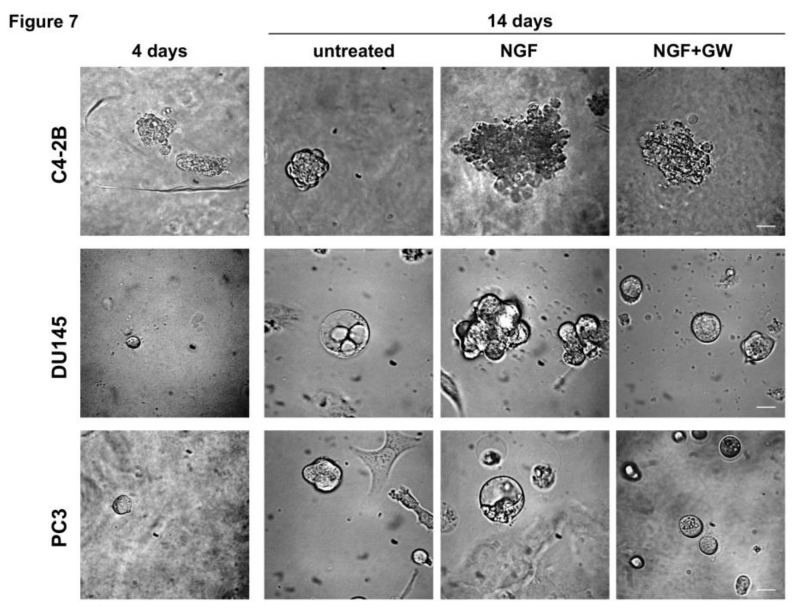
Nerve growth factor (NGF) increases the growth of organoids from castration resistant- prostate cancer (CRPC) cells through Tropomyosin receptor kinase (TrkA) activation. C4-2B (upper panel), DU145 (middle panel) and PC3 (lower panel) were used in miniaturized 3D cultures in extracellular matrix (ECM), as reported in Methods. Four days after cells embedding in VitroGel-3D-RGD, representative images were acquired as described in Methods. 3D cultures were left untreated or treated with 100 ng/mL NGF, in the absence or presence of GW441756 (GW; 1 μM) for 14 days. Shown are phase-contrast images captured at 14th day. Scale bar, 100 µ.

**Figure 8 cancers-11-00784-f008:**
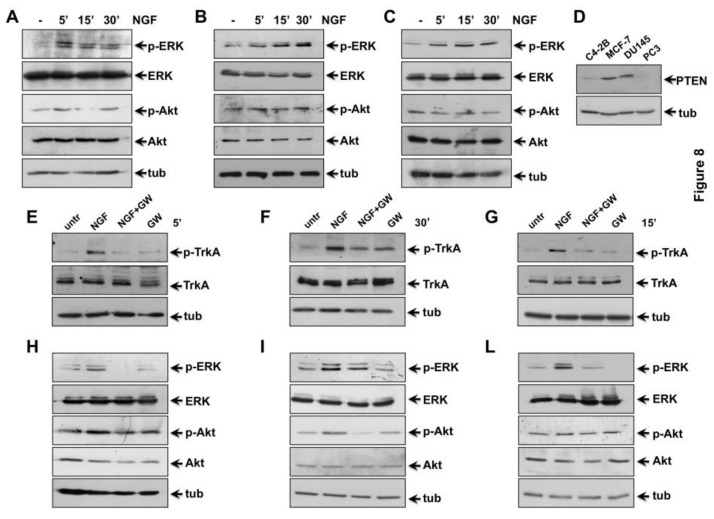
Tropomyosin receptor kinase A (TrkA) activation by Nerve growth factor (NGF) triggers mitogen-activated protein kinases (MAPKs) and Akt activation in castration-resistant prostate cancer (CRPC) cells. Quiescent C4-2B (**A**, **E** and **H**), DU145 (**B**, **F** and **I**) and PC3 (**C**, **G** and **L**) cells were used. In **A**–**C**, cells were left un-stimulated or stimulated for the indicated times with 100 ng/mL NGF. In **D**, the indicated cycling cells were used. In **E**–**L**, cells were left un-stimulated or stimulated for the indicated times with 100 ng/mL NGF, in the absence or presence of GW441756 (GW; 1μM). In **A-C,** protein lysates were analyzed by western blot, using the indicated antibodies. Filters were re-probed using anti-ERK (Extracellular signal-regulated kinase) or anti-Akt or anti-tubulin (tub) antibodies, as a loading control. In (**D**), protein lysates were analyzed by Western blot, using the indicated antibodies. In **E**–**F**, protein lysates were analyzed by western blot, using the antibodies against the indicated proteins (p-TrkA stands for P-Tyr-490 TrkA). In **H**–**L**, cells were left untreated (untr) or treated for the indicated times with NGF (100 ng/mL), in the absence or presence of 1μM GW441756 as reported in **E**–**G**. Lysates proteins were analyzed by western blot, using the antibodies against the indicated proteins. In **A**, **B**, **C**, **H**, **I**, **L**, p-ERK stands for P-Tyr 204 ERK 1, and the corresponding phosphorylated ERK 2; p-Akt stands for P-Ser 473 Akt. The filters were re-probed using anti-ERK or anti-Akt or anti-tubulin (tub) antibodies, as loading controls.

**Figure 9 cancers-11-00784-f009:**
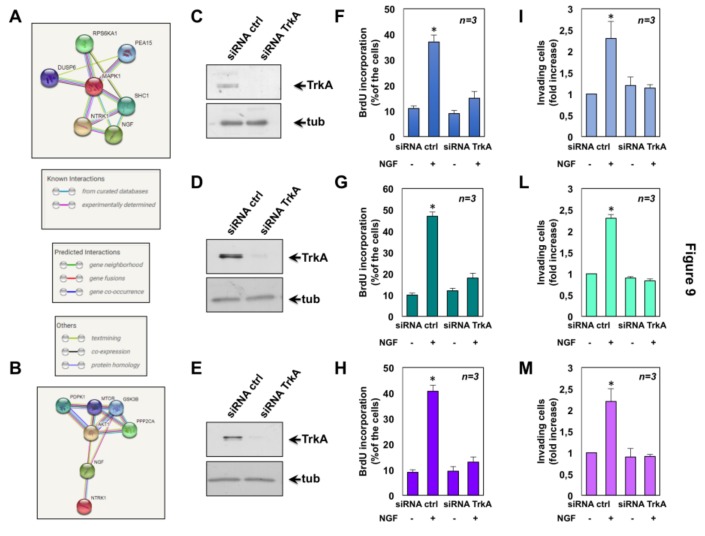
Results from Search Tool for the Retrieval of Interacting Genes/Proteins (STRING) network and effect of Tropomyosin receptor kinase A (TrkA) knockdown in castration-resistant prostate cancer (CRPC) cells. Combined screenshots from the STRING website, which has been queried with TrkA (NTRK1) and extracellular signal-regulated kinase (ERK; MAPK1; **A**) or TrkA (NTRK1; neurotrophic tyrosine kinase) and Akt (or PKB, protein kinase B, AKT1; **B**) are shown. Colored lines between the proteins indicate the various type of interaction, as described in the inset in **A**. Protein nodes also indicate the availability of 3D protein structure information. C4-2B (**C**), DU145 (**D**) and PC3 (**E**) cells were transfected with TrkA siRNA or control siRNA, as reported in Methods. After transfection, the cells were made quiescent and then left un-stimulated or stimulated with Nerve growth factor (NGF). In **C**–**E**, protein lysates were analyzed by western blot using the antibodies against the indicated proteins. Transfected C4-2B (**F**), DU145 (**G**) and PC3 (**H**) cells on coverslips were left untreated or treated for 18h with 100ng/mL NGF. After in vivo pulse with 100 μM BrdU, BrdU incorporation was analyzed by IF and expressed as % of total cells. Transfected C4-2B (**I**), DU145 (**L**) and PC3 (**M**) cells were used for invasion assays in Boyden’s chambers pre-coated with Matrigel. NGF (100 ng/mL) was added to the upper and the lower chambers. After 24 h, invading cells were counted as reported in Methods. In **F**–**M**, the cells were co-transfected with siRNA Alexa Fluor 488 to help identification of transfected cells. Results from three different experiments were collected and expressed as fold increase. Means and standard error of the means (SEMs) are shown. *n* represents the number of experiments. * *p* < 0.05 for the indicated experimental points vs. the corresponding untreated control.

**Figure 10 cancers-11-00784-f010:**
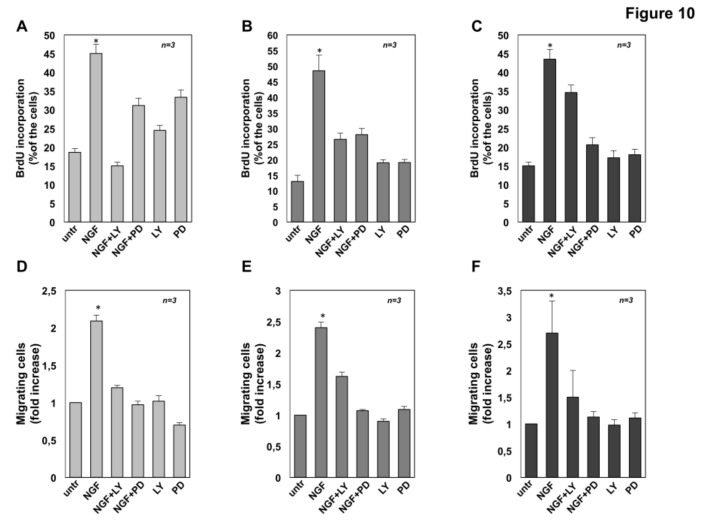
Inhibition of Akt or extracellular signal-regulated kinase (ERK) impairs the nerve growth factor (NGF)-challenged proliferation and migration in castration resistant prostate cancer (CRPC) cells. C4-2B (**A**), DU145 (**B**) and PC3 (**C**) cells were left untreated (untr) or treated for 18h with the indicated compounds. After in vivo pulse with 100 μM BrdU, BrdU incorporation was analyzed by IF and expressed as % of total cells. C4-2B (**D**), DU145 (**E**) and PC3 (**F**) cells were used for migration assay in collagen pre-coated Boyden’s chambers. The indicated compounds were added to the upper and the lower chambers. After 7 h, migrated cells were counted as described in Methods and results expressed as fold increase. In **A-F**, NGF was used at 100 ng/mL; PD98059 (PD) and LY-294002 (LY) were used at 10 μM. Results from three different experiments have been collected. Means and standard error of the means (SEMs ) are shown. *n* represents the number of experiments. * *p* < 0.05 for the indicated experimental points vs. the corresponding untreated control.

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
