# Peer review of "Nerve Growth Factor Induces Proliferation and Aggressiveness in Prostate Cancer Cells"

_cancers, 2019, doi:10.3390/cancers11060784_

Round 1

Reviewer 1 Report

The authors studied NGF role on proliferation and migration/invasion in castration-resistant prostate cancer cells and clarified its mechanisms. NGF activated ERK and partially Akt through TrkA phosphorylation. PI3K inhibitor reduced NGF-induced proliferation and migration, supporting involvement of ERK in NGF-TrkA induced progression of prostate cancer cells. These results seem to be novel and important findings; however, there may be several points to be amended and illustrated in detail.

Major:

1.       There are inconsistent results among 3 cell lines, even authors explain at each point, indicating weakness of conclusion of this study.

Minor:

1.       In Figure 1, the increase ratios of cell proliferation treated with NGF and serum are almost similar in 3 cell lines. It may be needed to explain what this phenomenon mean, that is, whether NGF is single contributor to cell proliferation in serum or not.

2.       It may be better to add “serum + GW” to Figure 2-4.

3.       The difference of cell morphology changes between untreated and NGF in Figure 5B and C is unclear, better pictures may be replaced.

4.       Figure 8 might be redundant because Figure 7 also includes similar results. These two figures might be rearranged.

5.       In Result, 2.8. line 9-12, if authors insist so, reference or data are needed.

6.       The concentration of NGF in tumor microenvironment should be written. It may be good to discuss whether NGF concentration (100 ng/mL) is appropriate (or not).

7.       The relationship between TrkA and PTEN might be described.

Author Response

Major:

      There are inconsistent results among 3 cell lines, even authors explain at each point, indicating weakness of conclusion of this study.

- We thank the Reviewer for this helpful comment. Therefore, we have strengthened our findings by adding new experimental approaches. Specifically, we have now shown by siRNA approach that TrkA is responsible for the NGF-elicited effects in all the CRPC cell lines employed (see the new Figure 9). Additionally, we have demonstrated by AR knockdown that NGF action in C4-2B cells is not related to AR, but additional characteristics of these cells, including: 

a) the presence of p75 (NTR), which acts as a tumor suppressor gene, in C4-2B, but not in DU145 and PC3 cells (Fig. 6 and last paragraph in section 2.3 of Results);

b) the difference in the expression levels of proteins involved in the basic machinery leading to cell motility and invasiveness (e.g, filamin A, FAK and paxillin). In this regard, see the data in the new Fig. 6 and last paragraph in section 2.3 of Results). See also the Discussion section at page 15 , lines 495-499.                   .

At last, the differences between the three cell lines here employed can be due to  the presence of AR variants or expression of the ubiquitin ligase TRAF4. In this regard, see also the Discussion section on page 15, lines 470-474  and page 15, lines 512-520.

 Minor:

1. In Figure 1, the increase ratios of cell proliferation treated with NGF and serum are almost similar in 3 cell lines. It may be needed to explain what this phenomenon mean, that is, whether NGF is single contributor to cell proliferation in serum or not.

- We agree with the Referee's comment. Therefore, we have included in the legend to Figure 1 data showing that serum-induced BrdU incorporation in C4-2B, DU145 and PC3 cells is unaffected by GW441756. Therefore, on the basis of these and other (Fig. 1 E-G) results, we can conclude that other growth factors present in serum are responsible for the observed effects. In this regard, see also paragraph 2.1 and the related new Ref. (Wight, 2000).

2.       It may be better to add “serum + GW” to Figure 2-4.

- Here again, GW441756 does not affect the motility and invasiveness induced by serum in CRPC cells. In this regard, see the new Fig.1S and paragraph 2.2.

3.       The difference of cell morphology changes between untreated and NGF in Figure 5B and C is unclear, better pictures may be replaced.

- We agree with the Referee's comment. Therefore, we have performed new experiments and captured  images at high magnification (see the new panels B and C in Fig. 5).

4.       Figure 8 might be redundant because Figure 7 also includes similar results. These two figures might be rearranged.

- We have consistently embedded the two Figures in the new Figure 8, and rearranged the relative results in one paragraph.

5.       In Result, 2.8. line 9-12, if authors insist so, reference or data are needed.

- We have added the reference for this paradoxical effect of MAPK inhibitors. In this regard, see the new section 2.8  in Results (page 13, lines 433-437).

6.       The concentration of NGF in tumor microenvironment should be written. It may be good to discuss whether NGF concentration (100 ng/mL) is appropriate (or not).

- This issue has been addressed in the Discussion section on page 14, lines 464-466 and page 15, lines 476-477.

7.       The relationship between TrkA and PTEN might be described.

-In the paragraph 2.6 (page 11, lines 357-360), it has been explained the link between TrkA, PI3-K signaling and PTEN. The appropriate Reference has been included.

Reviewer 2 Report

The manuscript by Marzia Di Donato et al provides information about the correlation of the activation of NGF-TrkA axis with proliferation, migration, and invasiveness in prostate cancer cells. They further showed that treated cells with NGF sensitizes the AR-negative prostate cancer cells underwent EMT, while increased NGF treatment does not seem to induce the EMT of AR-positive cell line. While the implication of NGF in prostate cancer progression is not entirely novel, in addition, NGF triggers MAPK and Akt activation in prostate cancer cells has been well studied. Data partially supporting the role of NGF in prostate cancer comes from various cells, even if not very novel. NGF certainly play roles in prostate cancer development and progression, but the mechanisms remain unknown.

Major Comments

One key information not provided by the present work is why NGF promotes aggressiveness of AR-negative cells through TrkA but this is not the case in AR-positive cells. The authors’ previous work pointed out a cross-talk between AR and TrkA in prostate cancer cells (Cell Death Discov. 2018; 4: 5). If AR expression is the root cause of migratory phenotype in C4-2B cells via interact with TrkA as the authors indicated, then does AR-negative cells undergo the same mechanisms to promote malignant progression? Or more importantly, does silencing of AR in C4-2B cells significantly modify the NGF effect on the aggressiveness? The authors have to provides this evidence. If it is connected to AR expression and signaling, how does NGF regulate AR? Without such information, it remains a correlation and the discovery only provides an incremental advance of our understanding.

As the title indicated, the major conclusion of this paper is that the induction of NGF treatment promotes malignancies. TrkA is but one component of NGF action pathway. Inhibition of other NGF receptors is important to reach this conclusion. In addition, it is import to use shRNA for NGF and TrkA knockdown (instead of simply using one TrkA inhibitor) to more convincingly rule out off side effects of drugs. Moreover, there have been several studies implicating NGF in CRPC progression (e.g., J Clin Invest. 2018 Jul 2;128(7):3129-3143). This paper focuses on TrkA regulates prostate cancer metastasis. The authors should discuss the present finding in the context of previous work.

Author Response

'One key information not provided by the present work is why NGF promotes aggressiveness of AR-negative cells through TrkA but this is not the case in AR-positive cells. The authors’ previous work pointed out a cross-talk between AR and TrkA in prostate cancer cells (Cell Death Discov. 2018; 4: 5). If AR expression is the root cause of migratory phenotype in C4-2B cells via interact with TrkA as the authors indicated, then does AR-negative cells undergo the same mechanisms to promote malignant progression? Or more importantly, does silencing of AR in C4-2B cells significantly modify the NGF effect on the aggressiveness? The authors have to provides this evidence. If it is connected to AR expression and signaling, how does NGF regulate AR? Without such information, it remains a correlation and the discovery only provides an incremental advance of our understanding'.

- We thank the Reviewer for this intriguing comment. In the revised version of the manuscript we have tried our best to meet the referee's concerns.

1) In the new Figure 5, we have silenced AR in C4-2B cells and we have shown that NGF action is not related to AR in C4-2B cells. These cells, indeed, still proliferate and migrate in response to NGF, even when AR is knockdown. Our data support the conclusion that, irrespective of AR, NGF promotes mitogenesis and invasiveness of C4-2B cells.

2) LNCaP and C4-2B cells exhibit, however,  important differences, including the number and distribution of mutations as well as gene expression (Spans et al., 2014). As extensively discussed in the revised paper, these differences might be crucial in dictating the choice of AR partners. Thus, in LNCaP cells, AR might easily associate with growth factor or neutrotrophin receptors (e.g., EGF-R or TrkA ), while in C4-2B cells, the receptor might recruit other signaling effectors (e.g., focal adhesion kinase and/or proteins involved in integrin pathway). In this regard, see the discussion section on page 15, lines 507-512.

3) C4-2B cells express AR variants that might modulate the NGF signaling. AR8 variant, for instance, controls the EGF signaling in C4-2B cells (Yang et al., 2011). Expression of this variant might negatively impact the responsiveness to NGF and the AR/TrkA crosstalk in C4-2B cells. These findings have been extensively discussed on page 15, lines 470-474 and page 15, lines 512-515.                                 

4) The possibility that differences in TRAF4 levels play a role in modulating the AR/TrkA cross talk in C4-2B cells as well as the strenght and duration of NGF-induced TrkA activation among the various CRPC cells cannot be excluded. These findings have been extensively discussed on page 15, lines 515-520.

5) The basic machinery leading to proliferation or migration is quite different in LNCaP cells, as compared with C4-2B cells. LNCaP cells, indeed, respond to androgens, while C4-2B are insensitive to the hormone in terms of both proliferation and migration (unpublished results from our lab). In our previous findings (Di Donato et al., 2018) we detected a bi-directional cross talk between AR and TrkA in LNCaP cells. Casodex inhibited, indeed, the NGF action, while GW inhibited the androgen effects. This cross talk cannot be recapitulated in C4-2B cells, because they are androgen-insensitive. That's way, we did not extensively study so far the cross-talk between AR and TrkA in C4-2B cells.

In sum, whatever the mechanism, our present data show that full-length AR is dispensable for NGF action in CRPC cells. 

'.........As the title indicated, the major conclusion of this paper is that the induction of NGF treatment promotes malignancies. TrkA is but one component of NGF action pathway. Inhibition of other NGF receptors is important to reach this conclusion. In addition, it is import to use shRNA for NGF and TrkA knockdown (instead of simply using one TrkA inhibitor) to more convincingly rule out off side effects of drugs....'

- We agree with the Referee's comment. Therefore, we have performed the TrkA silencing in CRPC cells. The data shown in the new Fig. 9 demonstrate that NGF signaling requires TrkA in the cell lines employed. In this regard, see also the new paragraph 2.7 in the Result's section. 

'.......Moreover, there have been several studies implicating NGF in CRPC progression (e.g., J Clin Invest. 2018 Jul 2;128(7):3129-3143). This paper focuses on TrkA regulates prostate cancer metastasis. The authors should discuss the present finding in the context of previous work...'

- We thank the Reviewer for this helpful comment. In the revised version of the manuscript, we have extensively discussed our data in the context of previous work in CRPC and the putative role of TRAF4 in modulating the NGF signaling in various CRPC cells has been discussed (page 15, lines            515-520).

Reviewer 3 Report

In this manuscript, Dr. Di Donato and coauthors build on their 2018 Cell Death Discov paper where NGF/NTRK1 signaling axis crosstalks with AR in LNCaP cells. Herein they use distant mCRPC models from bone (LNCaP-C4-2B, PC3) and brain (DU-145) to demonstrate NGF/NTRK1 signaling mediates cell proliferation, invasiveness and EMT. Notably, they use organoid cultures as well as cell culture to demonstrate increased proliferation after treatment with NGF. The NGF signaling axis is inhibited in all models by GW441756. They demonstrate that the signaling cascade is thorough MAPK and Act. NGF/NTRK1 signaling also induced EMT in models except C4-2B. The data supports NGF/NTRK1 signaling PCa progression and shows that inhibition of this axis controls cell proliferation and migration. The manuscript is well referenced and statistics used are appropriate.

I suggest:

1) spell check lines 206 and 379. Also check the figure title bold font.

2) authors discuss how their data can improve NGF/NTRK1 PCa clinical trials (lines 415-419).

3) comment on AR-variant expression in the models used and possible impact of variant AR signaling on NGF/NTRK1 axis.

Author Response

In this manuscript, Dr. Di Donato and coauthors build on their 2018 Cell Death Discov paper where NGF/NTRK1 signaling axis crosstalks with AR in LNCaP cells. Herein they use distant mCRPC models from bone (LNCaP-C4-2B, PC3) and brain (DU-145) to demonstrate NGF/NTRK1 signaling mediates cell proliferation, invasiveness and EMT. Notably, they use organoid cultures as well as cell culture to demonstrate increased proliferation after treatment with NGF. The NGF signaling axis is inhibited in all models by GW441756. They demonstrate that the signaling cascade is thorough MAPK and Act. NGF/NTRK1 signaling also induced EMT in models except C4-2B. The data supports NGF/NTRK1 signaling PCa progression and shows that inhibition of this axis controls cell proliferation and migration. The manuscript is well referenced and statistics used are appropriate.

I suggest:

1) spell check lines 206 and 379. Also check the figure title bold font.

- Consistent with the Referee's comments, we have checked the indicated lines together with the Figure title bond font.

2) authors discuss how their data can improve NGF/NTRK1 PCa clinical trials (lines 415-419).

-  We thank the Referee for this exciting comment. Therefore, we have discussed the therapeutic implications of our findings and we have placed our data into a more general context of drug-resistance of PC. In this regard, see the Discussion section on page 16, lines 530-546.                . 

3) comment on AR-variant expression in the models used and possible impact of variant AR signaling on NGF/NTRK1 axis.

- Here again, we thank the Reviewer for this intriguing comment. We have now extensively discussed the impact of AR variants in the observed findings.  In this regard, see the Discussion section on page          15, lines 470-474 and  lines 512-515.                 

Round 2

Reviewer 1 Report

The manuscript was amended nicely.

Author Response

Dear Reviewer,

Many thanks for your reply and especially for your previous comments.

Best regards,

Gabriella Castoria